# Morphogenesis Dynamics in Leishmania Differentiation

**DOI:** 10.3390/pathogens11090952

**Published:** 2022-08-23

**Authors:** Ramu Dandugudumula, Renana Fischer-Weinberger, Dan Zilberstein

**Affiliations:** Faculty of Biology, Technion—Israel Institute of Technology, Haifa 3200003, Israel

**Keywords:** leishmania, differentiation, subpellicular microtubules, morphogenesis, image streaming

## Abstract

Leishmania, the causative agent of leishmaniasis, is an obligatory intracellular parasite that cycles between phagolysosome of mammalian macrophages, where it resides as round intracellular amastigotes, and the midgut of female sandflies, where it resides as extracellular elongated promastigotes. This protozoan parasite cytoskeleton is composed of stable and abundant subpellicular microtubules (SPMT). This study aims to determine the kinetics of developmental morphogenesis and assess whether microtubules remodelling is involved in this process. Using image-streaming technology, we observed that rounding of promastigotes during differentiation into amastigotes was initiated promptly after exposure to the differentiation signal. Stabilizing microtubules with taxol sped rounding, but later killed differentiating parasites if taxol was not removed. Microtubule destabilizers such as vinblastine had no effect on the rate of rounding, nor on the viability of differentiating parasites. In the reverse process, elongation is initiated after a delay of 7.5 and completed 72 h after exposure to the amastigote to the promastigote differentiation signal. During the delay, parasites became highly sensitive to treatment with microtubule destabilizers. The addition of vinblastine during the first 7.5 h halted differentiation and killed parasites. Between 8 and 24 h, parasites gradually became resistant to vinblastine and, in parallel, started to elongate. In contrast, taxol had no effect on parasite elongation, nor on the viability of these cells. In a parallel study, we showed that the Leishmania-specific protein kinase A (PKA) holoenzyme containing the LdPKAR3-C3 complex is essential for promastigote elongation. Mutant promastigotes lacking either of these proteins are round, but maintain their flagella. Here, we observed that during differentiation into amastigotes, these mutants round at the same rate as the wild type, but never exceed the WT density of round amastigotes. In the reverse process, these mutants undergo the same initial delay and then elongate at the same rate as the WT. They stop elongating when they reach 20% of elongated cells in mature promastigotes. Our analysis indicates that while promastigote rounding into amastigotes did not require microtubule remodelling, morphogenesis of round amastigotes into elongated promastigotes required microtubule rearrangement before elongation was initiated. This is the first study that investigates the dynamics of microtubules during parasite development.

## 1. Introduction

Leishmaniasis, caused by protozoan parasites of the genus *Leishmania*, is a debilitating disease, and is a major health burden globally due to its high morbidity and low mortality [1,2]. This organism exhibits a digenetic life cycle that shuttles between the midgut of sandflies and the phagolysosomes of mammalian macrophages [3,4]. *Leishmania* undergoes a series of morphological changes as they cycle through its vectors and hosts. Parasites are found as promastigotes within the insect, defined by an elongated cell body with a long motile flagellum. Conversely, *Leishmania* enters the mammalian host cells and differentiates into amastigote, characterized by a spherical cell body with a short immotile flagellum [4,5,6]. This utterly distinct and characteristic morphology in vector and host, which changes dynamically during differentiation of *Leishmania*, is elusive, yet fascinating. Cell morphology is regarded as an important hallmark of differentiation in eukaryotic cells [7]. Therefore, characterization of cell morphogenesis aids in developing a better understanding of the differentiation.

The basic scaffold that determines the shape of trypanosomatids, including *Leishmania*, is composed of stable microtubules that run under the pellicle, namely the subpellicular microtubules (SPMT). These SPMT of trypanosomatid cytoskeleton are extensively cross-linked to each other and to the plasma membrane [8,9,10]. Interestingly, while the long motile flagellum of promastigote has the canonical 9 + 2 axoneme, the short nonmotile flagellum of amastigotes has a 9 + 0 (9v) axoneme arrangement [11,12]. Interestingly, differentiation-induced flagella shortening and rearrangement is independent of cell morphology [11]. Hence, limited information is available on the developmental morphogenesis of *Leishmania*.

The development of a host-free differentiation system that mimics the intra-lysosomal environment has enabled a better understanding of the molecular mechanism of *Leishmania* development [13,14]. Based on confocal microscopy, Barak et al., determined that promastigotes start to round at 5 h (phase II) and the process completes at 24 h (phase III) of post differentiation signal. In the reverse direction, amastigotes’ elongation to promastigotes begins at around six hours of differentiation and takes an additional 48 h to complete [15]. To date, except for flagella biogenesis, no quantitative analyses of the developmental morphology of the *Leishmania* cell body have been investigated.

In the present work, we employed an image stream flow cytometer to quantitatively determine the kinetics of *Leishmania* cell morphogenesis during differentiation. The shape of cells was determined by calculating the aspect ratio (width divided by length) of each cell at each time point of differentiation. The analyses indicated that rounding of elongated promastigote into amastigote-shaped cells initiates promptly after exposure to differentiation signal. Conversely, amastigote elongation initiates only 7.5 h after exposing mature amastigotes to promastigote differentiation signal. Using microtubule-specific drugs, we observed that this delay was required for microtubule rearrangement, a process that was not required for promastigote rounding. Both processes required microtubule stability.

Our findings indicate the significance of SPMT remodelling for the amastigote morphogenesis to promastigote, specifically in the initial hours of differentiation. This study provides concrete evidence of the role of microtubule remodelling involved in the *Leishmania* morphogenesis.

## 2. Materials and Methods

### 2.1. Chemicals

Paclitaxel (Taxol) and vinblastine sulphate salt were obtained from Sigma-Aldrich, Burlington, MA, USA. All other materials used in this study were of analytical grade and commercially available.

### 2.2. Leishmania Cell Culture and Differentiation

Promastigotes were grown in medium 199 (Sigma-Aldrich, Rockville, MD, USA) at 26 °C, supplemented with 10% heat-inactivated foetal bovine serum (FBS; Biowest) and 100 units/mL penicillin (Sigma-Aldrich, Burlington, MA, USA), 100 μg/mL streptomycin (Sigma-Aldrich, St. Louis, MO, USA). Promastigote cultures of the *L. donovani* Bob strain (LdBob strain/MHOM/SD/62/1SCL2D), initially obtained from Dr Stephen Beverley (Washington University, St. Louis, MO, USA), and *L. donovani* 1S strain, (MHOM/SD/00/1S) were used in this study. 

Axenic differentiation of promastigotes to amastigotes was carried out as decried previously [14]. Briefly, late log phase promastigotes were inoculated in medium 199 at pH 5.5 containing 25% foetal calf serum, and incubated at 37 °C, 5% CO_2_. The differentiation of axenic amastigotes to promastigotes was carried out, as described in Bachmaier et al. [15]. Briefly, log phase axenic amastigotes (0.5 × 10^7^ cells/mL) were taken and centrifuged at 1244× *g* for 10 min (room temperature). Further, amastigotes were resuspended in promastigote medium (M199 at pH 7 + 10% FCS) to a density of 0.5 × 10^6^ cells/mL and incubated at 26 °C. At each time point, parasites were collected, washed in 1× PBS and used for microscopy and lysate sampling.

### 2.3. Cell Viability Assay 

The cytotoxic effect of compounds was evaluated based on cleavage of MTT ((3-[4,5- dimethylthiazol-2-yl]-2,5-diphenyl tetrazolium bromide) by viable cells into formazan crystals. Briefly, parasites were washed twice with 1× PBS and suspended into a 96-well microtiter plate at final volume of 100 µL supplemented with 5 mM glucose (100 µL/well with). MTT [3-(4,5-dimethyl-2-thizolyl)-2,5-diphenyltetrazolium bromide] dye solution (Sigma-Aldrich, Rockville, MD, USA) (5 mg MTT in 1 mL PBS) was added at 100µg of final concentration and incubated at 37 °C in a 5% CO_2_-air atmosphere for 3 h. Parasites with amphotericin B (2.5 µg/mL) (Sigma-Aldrich, Rockville, MD, USA) was maintained as the positive control. Formazan crystals formed after incubation were solubilized with 10% SDS and incubated at 37 °C for 30 min. The change in colour from yellow to purple was read at an absorbance of 570 nm. The results are calculated as the percentage of viability in relation to the untreated cells.

### 2.4. Image Stream Flow Cytometry Analysis

Image Stream analysis was performed to study the morphology of parasites. Parasites at their respective time points of differentiation were taken and washed with 1× PBS. Further, cells were fixed in 4% paraformaldehyde. The samples were run using the Imagestream Mark II, and Bright field images at 40× magnification were acquired (Amnis). The data were analysed with IDEAS software, version 6.10, with compensation according to the software standards. Cells were gated for single cells in focus. Further, cells were classified based on shape (Cylindrical (elongated), Ovoid, Round) by considering aspect ratio vs. symmetry features. The aspect ratio feature assesses the elongation of cells by height/width ratio. Symmetry 2 measures the tendency of the object to have a single axis of elongation.

### 2.5. Isolation of Knockout Mutants ∆ldpkar3 and ∆ldpkac3

*L. donovani* promastigote lines with deletion of the genes encoding LdPKAR3 and LdPKAC3 by targeted homologous recombination are described in Fischer-Weinberger et al. [16].

## 3. Results

### 3.1. Differentiation-Derived Parasite Rounding and Elongating Process Differences

Herein, we used a host-free differentiation system combined with image stream flow cytometry analysis to establish the morphology kinetics during the time course of differentiation. Promastigotes and amastigotes were exposed to their corresponding differentiation signals and aliquots fixed at different time points throughout the process. Subsequently, parasites were perceived though ImageStream^®^X Mk II, and 10,000 cells images per a group were acquired. Using the IDEAS software, the aspect ratio (AR), which is the width divided by its length, was determined for each cell [17,18]. As shown in Table 1, the mean AR of late log phase axenic promastigote population was 0.49, while that of amastigotes was 0.75. The software divided each culture into three AR groups (Figure 1); elongated with AR range of 0.1–0.4, ovoid with AR of 0.4–0.6 and round with AR of 0.6–1.0. Ovoid cells have an intermediate shape that is closer to round than to elongated (Appendix A). Interestingly, the majority (78.3%) of the mature amastigote population is round, whereas only 4.3% are elongated. In contrast, the promastigote population consists of only 58% elongated cells, and the portion of round cells is 17.1%. The round promastigotes have the typical flagella with the 9 + 2 axoneme, while the elongated amastigote cells are aflagellated. Fischer-Weinberger et al. [16] showed that LdPKAR3-C3 holoenzyme binding to the SPMT is necessary to maintain the elongated shape of promastigotes. Deleting either the LdPKAR3 or LdPKAC3 genes increased the proportion of round cells from 17% in wild type to 43% and 52% in the mutants, respectively. Cumulatively, this suggests that the round shape is a more relaxed than elongated shape for the SPMT cytoskeleton in *Leishmania*.

Next, we determined the change in the mean AR during promastigote differentiation into amastigotes (Figure 2A). As shown, promastigotes start to round immediately after exposure to the differentiation signal (phase I, [13]) and reach the maximum AR of 0.75 within 50 h, early in the process of amastigote maturation (phase IV). The same rate and graph shape were obtained when we plotted the relative increase in only the round cell population (Figure 2B), indicating that the rate of rounding only represents morphogenesis of the entire population. Promastigote mutants lacking the LdPKAR3 and LdPKAC3 genes were subjected to the differentiation signal to determine whether they further round despite the relatively high number of round promastigotes. As shown in Figure 2C, these mutant cells have an accelerated rate of rounding. They reached the same steady state of WT within 24 h, compared to 50 h in WT. Interestingly, rounding of both mutants levelled off at the same cell density as WT i.e., 78%. This indicates that regardless of the proportion of promastigote round cells, round amastigotes of LdPKAR3 or LdPKAC3 mutants will not exceed that of WT.

In contrast with the rapid process of promastigote rounding, following amastigotes exposure to differentiation signal, we observed a delay of 7.5 h during which no change in parasite shape was observed (Figure 3A,B). Cell elongation initiated at 8 h post exposure to the differentiation signal and was slower than the rate of rounding. It took about 72 h for parasites to elongate back to a mean AR of 0.5, which is typical for the promastigote population (Figure 2A,B). A mirror of the same graph was obtained when the percentage of elongated cells was plotted (compare Figure 3A,B). This further supports the idea that following the leading shape represents the morphogenesis of the entire population. Analysing the rate of elongation of the PKA mutants, Δ*ldpkar3* and Δ*lapkac3* indicated that they underwent the first delay of 7.5 h, as did WT. Subsequently, elongation initiated as in WT, but was limited to no more than 20% of the parasite population (Figure 3C). This indicates that lack of either LdPKAR3 or LdPKAC3 effects parasites’ ability to elongate, not the initial process that initiates elongation.

### 3.2. Microtubule Remodelling Is Required for Parasite Elongation, but Not for Rounding

Cell morphology, including that of *Leishmania*, is usually maintained by a cytoskeleton [19,20]. However, distinct from non-trypanosomatid Eukaryotes, the *Leishmania* cytoskeleton is dominantly composed of rigid SPMT [10,21]. In light of the kinetic differences between rounding and elongation, we directed our study to assess whether the cytoskeletal shape change is mediated through SPMT remodelling. This was achieved by employing two microtubule-specific drugs, taxol and vinblastine. Taxol induces microtubule polymerisation, thereby stabilizing the SPMT [8,22]. In contrast, vinblastine induces depolymerization of microtubules and thereby destabilizes the SPMT [23,24]. Both compounds have been used against *Leishmania* microtubules. Promastigotes were subjected to the differentiation signals in the presence of these microtubule active compounds, at concentrations previously used for *Leishmania* [25,26]. As shown in Figure 4A, treating differentiating promastigotes with 15 µM taxol was not toxic to cells within the first 24 h of differentiation. However, the parasites died 48 h post differentiation signal. In contrast, vinblastine had no effect on parasite viability nor differentiation. Taxol accelerated the rate of rounding exponentially where it reached saturation within 12 h of differentiation (Figure 4B and inset). In the presence of taxol, cells became round at five hours, much faster than the untreated cells (Appendix A). Conversely, vinblastine had no effect on rounding of parasites, and the rate of rounding was found to be similar to untreated parasites. Taken together, these results suggested that microtubule stability speeds up the rounding but became toxic when parasites entered the phase of amastigote maturation (phase IV). Furthermore, no SPMT remodelling was required to initiate the process of differentiation-induced rounding.

Unlike the process of promastigote rounding into amastigote-shaped cells, elongation during amastigote to promastigote differentiation started only 7.5 h after exposure to the differentiation signal (Figure 3). We hypothesized that this delay is required to enable SPMT rearrangement before they start elongating. To assess this hypothesis differentiating amastigotes were treated with taxol and vinblastine to assess their effect on viability and elongation kinetics. Late log phase amastigotes were transferred to promastigote medium at 26 °C to initiate their differentiation into promastigotes. At times indicated in closed circles in Figure 5A, 15 µM taxol (●), 25 µM vinblastine (●) or the same volume of sterile water were added (●). Incubation continued until 72 h after differentiation initiated. All cells were subjected to viability analyses using the MTT colorimetric method. As shown in Figure 5A, taxol has no effect on parasite viability, nor did this compound had any effect on the kinetics of elongation (Figure 5B). In contrast, adding vinblastine at the same concentration as in Figure 4 was cidal to differentiating parasites 0 to 7.5 h after the signal. These cells died and therefore were unable to differentiate into promastigotes. However, adding vinblastine at 8.5 h on, cells gradually developed resistance to vinblastine (Figure 5A red line and Figure 5C green and orange lines). Twenty-four hours after exposure to differentiation signal, parasites became completely insensitive to the compound (Figure 5C). Consistent with the development of resistance to vinblastine, the rate of parasite elongation increased. At 24 h post signal, the rate of elongation was identical to that of untreated cells (Figure 5C). 

In conclusion, these results suggested SPMTs remodelling at the initial hours of differentiation is crucial for amastigotes’ morphogenesis into the elongated promastigote. This rigorous remodelling at the initial hours is postulated as the cause of lag in the initiation of differentiation. Microtubule stability caused by Taxol has no role in the process of elongation, but is required for the rounding of parasites. 

## 4. Discussion

This study aimed to illustrate the kinetics of cell shape development during *Leishmania* differentiation, and the dynamics of the subpellicular microtubule (SPMT) cytoskeleton involved. The major observations are (1) elongated promastigotes start to round promptly immediately after exposure to differentiation signal, while round amastigotes start to elongate after a delay of 7.5 h, (2) SPMT stability speeds up rounding, and is essential for elongation initiation, and (3) sensitivity of differentiating amastigotes to the microtubule destabilizer vinblastine is transient, and parasites develop resistance to a drug that induces microtubule instability at >7.5 h of amastigote to promastigote differentiation.

Our results strongly support the idea that round is the prompt shape of *Leishmania* cells. Evidently, the mature amastigote population consists of 80% round but only 4% elongated cells, while the promastigote population consists of 60% elongated and 17% round cells. Support for this idea comes from previous observations indicating that metabolic starvation, as well as heat shock, caused promastigote populations to round [27,28,29]. The effect of temperature elevation on rounding was demonstrated in two independent cases. One elevating promastigote growth temperature to 37 °C induced *L. mexicana* promastigote to round to an amastigote-like shape. Joshi et al. [30] showed that shifting elongated promastigotes to 37 °C in a medium of 100% heat-inactivated serum induced elongated promastigotes to round to amastigote-shaped cells. Our recent observation that the presence of protein kinase A holoenzyme (LdPKAR3-C3) that anchors SPMT is essential for the elongated shape of promastigotes further supports this idea [16]. This observation is intriguing, as it is currently unclear whether the same applies to the other members of this family. The trypanosoma cruzi life cycle that includes intracellular amastigote also contains the *PKAR3* gene. In contrast, *African Trypanosomes* that are extracellular parasites with elongated shapes throughout their life cycle (TriTrypDB) lack this gene.

While our experiments clearly indicate that amastigote elongation initiates after a long delay, it is not clear what causes this delay. One reason could be the necessity of the presence of the PKA holoenzyme, i.e., the LdPKAR3-C3 heterodimer [16]. All catalytic subunits of PKA, including LdPKAC3 proteins, express only in promastigotes and are missing from the amastigote proteome [31]. However, here, we showed that a delay of 7.5 h exists in the mutants lacking either LdPKAR3 or LdPKAC3. However, they differ from WT in their limited ability to round. Both mutants were unable to exceed 20% elongated cells in the promastigote population. This phenomenon further supports our previous observation that the presence of PKA is influential only in promastigotes [15].

An interesting observation made in this study is that rounding of promastigotes lacking either LdPKAC3 or LdPKAR3 was, on the one hand, faster than WT, but could not exceed the density of the WT round cell population (i.e., 80%). Conversely, in the process of elongation, the mutant cells did not exceed the 20% of elongated cells in the promastigote population. This further supports the notion that the role of PKA in morphogenesis is limited to elongation. Does this mean that parasites obey a density constant that limits them to 80% round cells?

Using the axenic differentiation experimental system together with ImageStream flow cytometry analysis for morphometry enabled the first-time measurement of *Leishmania* shape dynamics during stage differentiation. The results of the new morphometric data presented pose interesting questions for future investigation into the molecular mechanisms relevant to the genes in the knockout mutants, as well as others yet to be identified that play a role in parasite morphogenesis.

## Figures and Tables

**Figure 1 pathogens-11-00952-f001:**
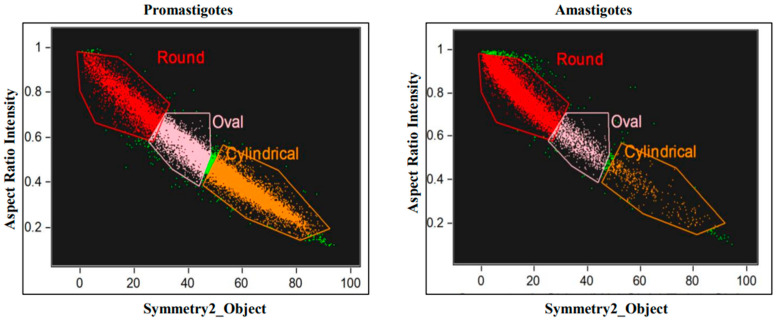
ImageStream morphology analysis of axenic *L. donovani* promastigotes and amastigotes. Late-log phase *L. donovani* promastigotes and amastigotes were subjected to ImageStream flow cytometry analysis as described in Materials and Methods. Parasites were classified into three different shapes (Cylindrical (elongated), Ovoid and Round) based on aspect ratio feature (AR) of width divided by length vs. symmetry 2 features.

**Figure 2 pathogens-11-00952-f002:**
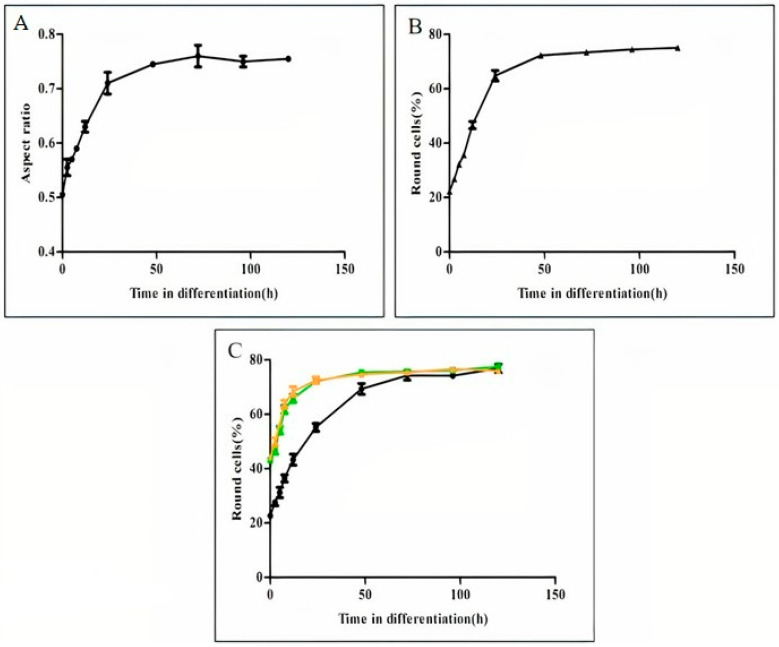
Time course analysis of rounding during promastigotes to amastigotes differentiation. Late log phases of wild type *L. donovani* promastigotes were subjected to differentiation into amastigotes, as described in Materials and Methods. At time points indicated on the graphs, 5 mL aliquots of cell suspension were subjected to ImageStream flow to determine the mean AR of 10,000 cells. (**A**) The graph illustrates the mean AR as function of differentiation progression, (**B**) percentage of cells in panel A with AR range of 0.6–1.0 (round) as function of differentiation progression, (**C**) the kinetics of cell rounding as function of differentiation progression of *L. donovani* mutants lacking either LdPKAR3 (■) or LdPKAC3 (▲) compared with wild type (●). Values are mean ± S.E.M. (Standard Error of the Mean, *n* = 3). At each time point, Image stream analyses 10,000 cells.

**Figure 3 pathogens-11-00952-f003:**
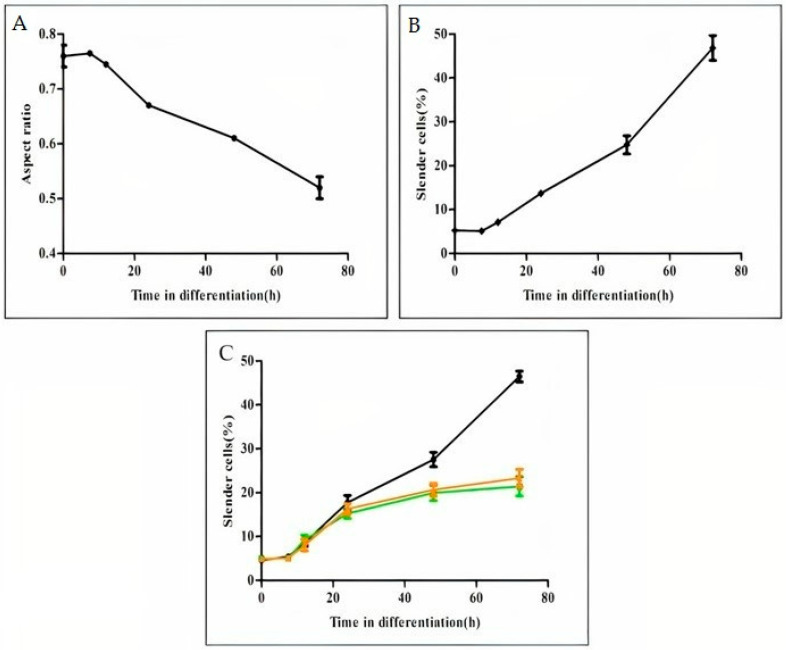
Time course analysis of cell elongation during amastigotes to promastigote differentiate log phase of wild type *L. donovani* amastigotes were subjected to differentiation into promastigotes, as described in Materials and Methods. At time points indicated on the graphs, 5 mL aliquots of cell suspension were subjected to ImageStream flow to determine the mean AR of 10,000 cells. (**A**) The graph illustrates the mean AR as function of differentiation progression, (**B**) percentage of cells in panel A with AR range of 0.1–0.4 (elongated) as function of differentiation progression, (**C**) the kinetic of cell elongation as function of differentiation progression of *L. donovani* mutants lacking either LdPKAR3 (■) or LdPKAC3 (▲) compared with wild type (●) during promastigotes to amastigote differentiation. Values are mean ± S.E.M. (*n* = 3). At each time point, Image stream analyses 10,000 cells.

**Figure 4 pathogens-11-00952-f004:**
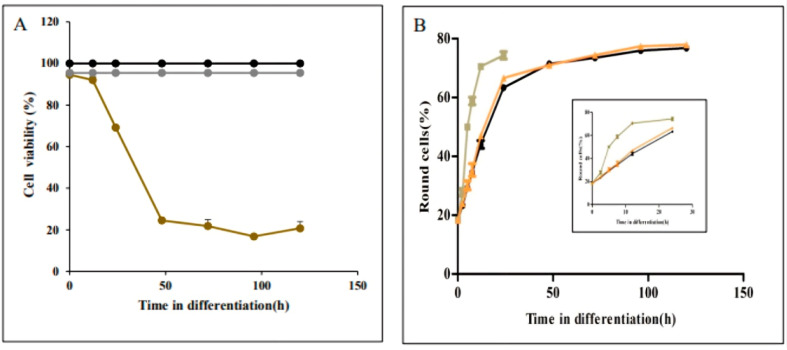
Late log phase of wild type *L. donovani* amastigotes were subjected to differentiation into promastigotes as described in materials and methods. (**A**) at time points indicated on the graph in close circles 17 µM taxol (●), 25 µM vinblastine (●) or same volume of sterile water (●) were added and cells continued to differentiate. When they reached 72 h aliquots were subjected for viability analyses using the MTT colorimetric assay. The graph illustrates per cent viable cells determined at 72h post signal. assess the cell viability during amastigotes to promastigotes differentiation. (**B**). Cells were exposed to differentiation signal in the absence (●) or presence of taxol (■) or vinblastine (▲). At each time point indicated on the graphs cells were subjected to image streaming analysis to determine percentage of elongated cells. Values are mean ± S.E.M. (*n* = 3). At each time point Image stream analysed 10,000 cells.

**Figure 5 pathogens-11-00952-f005:**
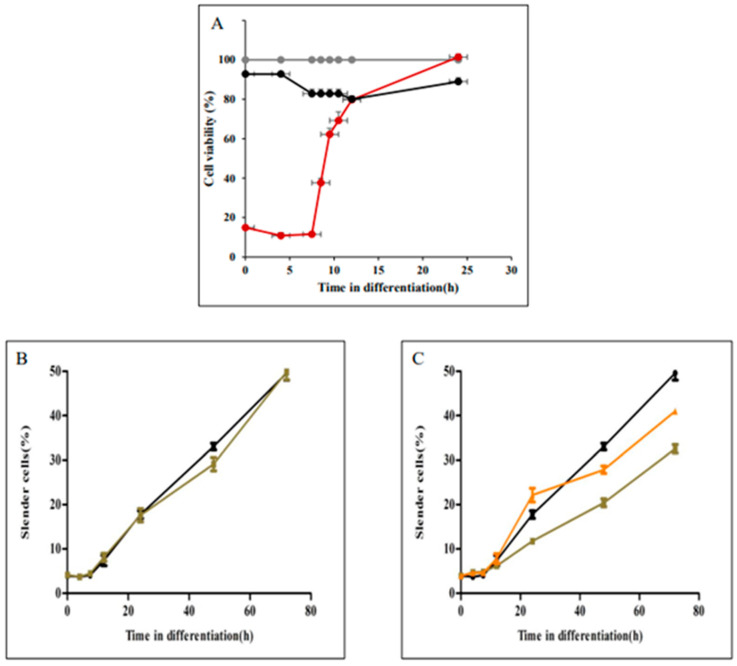
Evaluation of the effect of microtubule active drugs taxol and vinblastine on cell elongation during amastigotes morphogenesis to promastigotes. Late log phase of wild type *L. donovani* amastigotes were subjected to differentiation into promastigotes as described in materials and methods. (**A**) at time points indicated on the graph in close circles 17 µM taxol (●), 25 µM vinblastine (●) or the same volume of sterile water (●) were added and cells continued to differentiate. When they reached 72 h aliquots were subjected for viability analyses using the MTT colorimetric assay. The graph illustrates per cent viable cells determined at 72 h post signal assess the cell viability during amastigotes to promastigotes differentiation. Cells were exposed to differentiation signal in the absence or presence of taxol or vinblastine. At each time point indicated on the graphs cells were subjected to image streaming analysis to determine percentage of elongated cells (**B**) Untreated (●), Taxol (■): (**C**) Untreated (●), vinblastine-treated at 9h (▲), vinblastine-treated at 24 h (▲). Values are mean ± S.E.M. (*n* = 3). At each time point, Image stream analysed 10,000 cells.

**Table 1 pathogens-11-00952-t001:** The mean aspect ratios and percentage cell populations of different shapes of cells in promastigotes and amastigotes.

Life Stage	Shape	Mean Aspect Ratio	Portion in Population (%)
Promastigote	All	0.45	100
	Elongated	0.37	58.3
	Ovoid	0.55	24
	Round	0.75	17.2
Amastigote	All	0.75	100
	Elongate	0.37	4.3
	Ovoid	0.57	14.4
	Round	0.81	78.6

## Data Availability

Not applicable.

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
