# Peer review of "Morphogenesis Dynamics in Leishmania Differentiation"

_pathogens, 2022, doi:10.3390/pathogens11090952_

Round 1

Reviewer 1 Report

The manuscript “Microtubule dynamics in developmental morphogenesis of Leishmania” authored by Dandugudumula et al, is aimed to investigate the microtubule remodeling happening during the differentiation process of L. donovani. Though it is evident that Leishmania undergoes extensive morphological changes during their Life cycle, very limited information available on the molecular basis of morphogenic events. 

While the manuscript is well written, the experiments are executed well and the results are clearly presented, altogether the overall aim of the paper and the results are not enough of an advance to the existing knowledge in the field. The topic of the paper deserves a more comprehensive study. The experiments described in the article also lacks proper control, especially the promastigote to amastigote differentiation. 'Rounding' is a very common response shown by Leishmania species to any environmental stimuli not specifically to differentiation signals. Therefore, these experiments must be supported by data which shows changes in stage specific markers or more appropriately changes in the axoneme. Flagellar remodeling is also a significant part of morphogenesis in Leishmania. 

Overall, in my opinion, the experimental data presented in the article is not enough to justify publication in 'Pathogens'.

Author Response

  1. Reviewer #1 interpreted our aims as follows, “Dandugudumula et al, is aimed to investigate the microtubule remodelling happening during the differentiation process of donovani”. We are afraid that the reviewer misinterpreted our objectives. In both the abstract and introduction, we stated that “This study aims to determine the kinetics of developmental morphogenesis and assess whether microtubules remodelling is involved in this process”. This study does not aim to mechanistically describe microtubule remodelling, but rather to qualitatively determine the time course of differentiation-derived change in morphology and whether these changes require microtubule remodelling. Differentiation-derived morphogenesis of Leishmania has never been assessed before. We thought that the axenic differentiation system can help shed light on this process. Therefore, we decided to provide the public with the basic information on how morphogenesis progresses during promastigotes differentiate into amastigotes and vs. Because we found major differences in the kinetics between the two pathways, we assessed, using known microtubule drugs, whether any of these processes require microtubule rearrangements or remodelling. We are afraid that the title we gave to this study might be misleading and therefore we changed it to “Morphogenesis dynamics of Leishmania differentiation”. Of course, this study should trigger further investigation of the mechanism behind the process we describe. In our opinion, this study shed light on an unknown process and is therefore worthy of publication in this journal.
  2. The reviewer stated, “The topic of the paper deserves a more comprehensive study. As indicated above, we have provided sufficient data to describe the time course changes in Leishmania differentiation-induced morphogenesis.
  3. The reviewer stated, “'Rounding' is a very common response shown by Leishmania species to any environmental stimuli not specifically to differentiation signals. Therefore, these experiments must be supported by data which shows changes in stage-specific markers. As I am sure the reviewer knows that our laboratory has extensively characterized Leishmania differentiation, including the timing of changes in gene and protein expression at almost every minute of this process. We think that such a well-established experimental system can be used as a reference for differentiation analysis without repeating the experiments we have already carried out. Among the differentiation signal-derived processes we have shown that differentiation-derived rounding is a signal-activated process, not a stress response. The reviewer is welcome to read our papers from early 2000 where we showed that exposing parasites to acidic pH only or temperature only does not induce morphogenesis, only the full signal, proving that rounding here is a signal-derived process. A famous example is C. elegance whose time course of development has been extensively studied and thereafter the data used for all their developmental biology studies in C. elegance.
  4. Flagellar remodelling is also a significant part of morphogenesis in Leishmania. As stated in the Introduction, differentiation-derived flagella biogenesis has been extensively studied. We know that in promastigotes the axoneme has the 9+2 structure whereas in amastigotes flagella shortening also involves axoneme change to 9+0. We focus on body morphogenesis because nothing is known about this process.

Reviewer 2 Report

Major points that need to be addressed for 2nd round of revisions:

1)      There needs to be clear legends placed into Figures 2 through 5, indicating which lines are PKAR3 and C3 mutants and which lines are from promasitgote and amastigote samples that are incubated with taxol and vinblastine. As it stands, the data can be confusing to follow without those descriptors. For example, for “Figure 5B2” (Figure 5C), I am confused as to which line is what under what vinblastine treatment time addition.  Clear legends are needed for these data.

2)      The inset within Figure 4B can be its own graph, Figure 4C.

3)      The cell viability graph in Figure 5A might be better displayed as a bar graph since there are various timepoints.

4)      There are error (standard deviation/standard error?) bars for your various timepoints in your graphs, but there is no indication of how many experimental or technical replicates (is everything 10,000 images/samples taken?) these timepoints represent.  This should be clarified.

5)      There is no mention of the data collected with the PKAR3 and C3 mutants in the abstract, but has it been tested what the effects of PKAR3 and C3 mutants with taxol or vinblastine are on number of elongated cells in promastigotes ie does either compound change the number of 20% elongated cells in promastigotes or have any other appreciable morphological or growth effect on these cells?

Minor points:

“et al” should be italicized throughout the text.

The text and legend concerning Figure 5 should match up; Figure 5B1 is referred to as “Figure 5B” in the text, but B1 in the legend and graph and Figure5B2 is labeled as “Figure 5C” in the text, but B2 in the legend and graph.

Line 5: Change “Israe” to “Israel”

Line 15: Add comma to before “but”

Lines 27-28: “Development” should be dropped as a keyword or replaced with something more descriptive

Line 76: “Vinblastine” should not be capitalized.

Line 98: Remove “with” before the 1st period.

Line 100: “Amphotericin” should not be capitalized.

Line 113: Remove the space between “elongated ness” to yield “elongatedness”

Line 118: Insert a comma before “was”.

Line 145: Change “fig. 8in” to “Fig. 8 in”.

Line 154: Remove the extra period.

Line 169: Change “R3” to PKAR3” or indicate in the text that “R3” is an abbreviation for “PKAR3”.

Figure 3 and Figure 5: Should “slender cells” be “elongated cells”?

Remove extra “Figure 5” text on top of the Figure 5 panels.

Line 273: Remove the apostrophe from “amastigatoes” and replace phrase with “is crucial for the morphogenesis of amastigotes into the elongated…”.

Line 290: Merge reference numbers into [22-24].

Author Response

  1. We agree with the reviewer and apologize for the concise legends. We have revised the legends to figures 2-5. I hope they better describe the experiments, and the reviewer will find answers to all his queries.
  2. The inset in figure 4B is a zoom-in to a portion of the graph and therefore we think that it cannot justify an independent panel.
  3. As suggested by the reviewer, we tried replacing the graph with bars but thought that the graph better illustrates the process. We hope the reviewer will agree with us and let us use the graph.
  4. Again, we apologize for not including the statistics. We have added to the legends of figures 2-5 a statement on the standard error of the mean + number of biological repeats. I hope this clarifies the statistics and the values of the error bars.
  5. We thank the reviewer for pointing out the lack of description of analyses of PKAR3 and C3 mutants in Abstract. We added the following sentences: “In a parallel study, we showed that the Leishmania-specific protein kinase A (PKA) holoenzyme containing the PKAR3-C3 complex is essential for promastigote elongation. Mutant promastigotes lacking either of these proteins are round but maintain their flagella. Here, we observed that during differentiation into amastigotes these mutants round at the same rate as the wild type, but never exceed the WT density of round amastigotes. In the reverse process, these mutants undergo the same initial delay and then elongate at the same rate as the WT. They stop elongating when they reach 20% of elongated cells in mature promastigotes.”

Minor points

  1. Done
  2. We revised the section with the results of figure 5. We hope the experiments and results are clear.
  3. Done
  4. Done
  5. Development replaced by image streaming
  6. Done
  7. Done
  8. Done
  9. Done
  10. Done
  11. Done

Round 2

Reviewer 1 Report

1.     I thank the authors for considering the points raised in my previous review report. Placing the word ‘Microtubule dynamics’ especially in the beginning of the title broadens the scope of the study and thus become misleading. The current tile does more justice to the work described in the manuscript and appropriately match the target audience. It also nullifies some other concerns previously raised by me.

2.     I am aware of authors contributions in establishing axenic system as valuable tool for studying clinically relevant differentiation process of Leishmania spp. However, the parameters used for in vitro differentiation by different research groups have shown variations in differentiation signals and stages. The limitations of axenic differentiation when it comes to mimicking the intracellular environment has been subject of scientific discussion in the past. Evidence suggests that It is crucial to assess the differentiation process by proper controls beside morphological analysis (reviewed in Dias-Lopes et al, Axenic amastigotes of Leishmania species as a suitable model for in vitro studies. Acta Trop. 2021). So I strongly believe that using biomarkers to validate morphogenesis will not only improve the validity of the results but also add more biological significance to the study.  

Author Response

We fully agree with the reviewer on the diverse protocols for axenic differentiation for the various Leishmania species. The important and excellent review by Dias-Lopes et al. emphasizes this situation. We have discussed this as well in a recent method paper and suggested that all researchers use one common protocol (see reference 14). However, to ensure consistency, all our analyses of Leishmania differentiation have been carried out using the same strain of L. donovani. This enabled us to build a solid tower of data using a single protocol for axenic differentiation. The most characterized and consistent phenomenon is the four phases of differentiation we have described based on morphology, gene, and protein expression. When one reads our papers, he/she may use the published differentiation time course for process orientation. We are employing the same approach in this study.

In line with the reviewer’s concern and to avoid confusion we have indicated in the results section the phase number when we describe a process at a specific time of differentiation.

I hope that we have provided the reviewer with the appropriate answer and that the paper will now be approved for publication.

Reviewer 2 Report

I am satisfied with the changes.  

Author Response

We thank the reviewer